# First Evidence of Anti-Steatotic Action of Macrotympanain A1, an Amphibian Skin Peptide from *Odorrana macrotympana*

**DOI:** 10.3390/molecules27217417

**Published:** 2022-11-01

**Authors:** Ilaria Demori, Zeinab El Rashed, Giulia De Negri Atanasio, Alice Parodi, Enrico Millo, Annalisa Salis, Andrea Costa, Giacomo Rosa, Matteo Zanotti Russo, Sebastiano Salvidio, Katia Cortese, Elena Grasselli

**Affiliations:** 1Department of Earth, Environmental, and Life Sciences (DISTAV), University of Genoa, Corso Europa 26, 16132 Genoa, Italy; 2Inter-University Center for the Promotion of the 3Rs Principles in Teaching & Research (Centro 3R), 56122 Pisa, Italy; 3Department of Experimental Medicine (DIMES), Section of Biochemistry, University of Genoa, Viale Benedetto XV 1, 16132 Genoa, Italy; 4Angel Consulting, Via San Senatore 14, 20122 Milano, Italy; 5Department of Experimental Medicine (DIMES), Cellular Electron Microscopy Laboratory, University of Genoa, Via Antonio de Toni 14, 16132 Genoa, Italy

**Keywords:** anti-steatotic, amphibian skin peptides, triglycerides, liver, NAFLD, peroxisome proliferator-activated receptors (PPARs), perilipins (PLINs)

## Abstract

Many different amphibian skin peptides have been characterized and proven to exert various biological actions, such as wound-healing, immunomodulatory, anti-oxidant, anti-inflammatory and anti-diabetic effects. In this work, the possible anti-steatotic effect of macrotympanain A1 (MA1) (FLPGLECVW), a skin peptide isolated from the Chinese odorous frog *Odorrana macrotympana*, was investigated. We used a well-established in vitro model of hepatic steatosis, consisting of lipid-loaded rat hepatoma FaO cells. In this model, a 24 h treatment with 10 µg/mL MA1 exerted a significant anti-steatotic action, being able to reduce intracellular triglyceride content. Accordingly, the number and diameter of cytosolic lipid droplets (LDs) were reduced by peptide treatment. The expression of key genes of hepatic lipid metabolism, such as PPARs and PLINs, was measured by real-time qPCR. MA1 counteracted the fatty acid-induced upregulation of PPARγ expression and increased PLIN3 expression, suggesting a role in promoting lipophagy. The present data demonstrate for the first time a direct anti-steatotic effect of a peptide from amphibian skin secretion and pave the way to further studies on the use of amphibian peptides for beneficial actions against metabolic diseases.

## 1. Introduction 

The amphibian skin is a complex organ able to exert a multitude of vital functions related to respiration, osmoregulation, thermoregulation and defense against physical and biological environmental threats [1,2]. The importance of this organ is underlined by the presence of several glands (mucous and granular) able to maintain the integrity and humidity of the skin. Among these glands, granular glands can secrete several proteins, such as immunoglobulins and lysozymes, and several types of peptides, with the aim of protecting the skin from any kind of insult [3,4,5]. The secretion of granular gland content occurs through the contraction of surrounding myoepithelial cells, triggered by the release into the bloodstream of epinephrine and/or norepinephrine stimulated by any kind of stress [6]. Thus, amphibian skin peptides promote wound healing and display antimicrobial activity against bacterial, viral and fungal infections, which are causing a worldwide species decline and even local extinctions [2,7]. To render each species able to occupy a specific niche and thus increase its fitness in a given environment, each amphibian species produces its own specific set of peptides with well-defined sequences. To date, more than 2000 peptides from amphibian skin have been characterized [8], showing multiple biological activities besides anti-microbial activity, including anti-oxidant, anti-inflammatory, immunomodulatory, anti-tumor and anti-diabetic activities [8,9]. Such a wide array of bioactive molecules represents a huge opportunity to disclose new therapeutic tools for human health [10]. Particularly, anti-oxidant and anti-inflammatory actions could be exploited against the global burden of metabolic diseases, such as metabolic syndrome and liver diseases. In the present work, we consider the skin peptide macrotympanain A1 (MA1), which is secreted by the Chinese odorous frog *Odorrana macrotympana* [11]. The genus *Odorrana* (Amphibia, Anura, Ranidae) is endemic to East and Southeast Asia and includes approximately 53 species [12], each producing its own specific set of peptides. Yang et al. [11] purified 80 skin peptides from three different Chinese odorous frogs; among them, MA1 caught our attention due to its high scavenging activity against free radicals, coupled with a strong anti-inflammatory action. 

Oxidative stress and inflammation are among the “multiple hits” involved in the pathogenesis of nonalcoholic fatty liver disease (NAFLD) [13]. This liver disease is a global health burden affecting 25% of the population worldwide, becoming more and more prevalent also in children [14]. Despite being reversible in its early stages, NAFLD can worsen to steatohepatitis, fibrosis, cirrhosis and hepatocellular carcinoma. As no pharmacological treatment has yet been approved for NAFLD, there is an urgent need to find new preventive and therapeutic strategies, namely safe and effective drugs to be integrated with lifestyle modifications, the latter being essential for optimal metabolic homeostasis [15]. As a matter of fact, metabolic dysfunction-associated fatty liver disease (MAFLD) has recently been proposed as a novel term that better overarches the current knowledge of fatty liver diseases associated with metabolic dysfunction [16]. Indeed, insulin resistance (IR) is crucial for the development of fatty liver, due to the reduced insulin sensitivity, particularly in the hepatic and adipose tissue, which leads to enhanced free fatty acid delivery to the liver and increased stimulation of anabolic processes [17]. Of interest, among the most intriguing properties of amphibian skin peptides, there are their multiple anti-diabetic actions [18]. 

In this scenario, we employed a well-established in vitro model of fatty liver consisting in fat-enriched FaO rat hepatoma cells that can be treated with potential anti-steatotic compounds [19,20]. By using this model, we have been able to demonstrate for the first time a significant anti-steatotic effect of an amphibian peptide such as MA1. 

## 2. Results

### 2.1. MA1 Does Not Affect FaO Cell Viability

FaO rat hepatoma cells were cultured as described in Section 4. Briefly, steatosis was achieved by cell exposure to a mixture of oleate/palmitate (OP, 0.75 mM) for 3 h [20,21]. Thereafter, cells were incubated for 24 h either in a control medium (OP steatotic cells) or in the presence of MA1 at different concentrations (0.1, 1 and 10 μg/mL, corresponding to 0.9–9 μM). The concentration range was chosen on the basis of previous studies that investigated the insulinotropic properties of amphibian peptides using in vitro models. These studies demonstrated the ability of several amphibian peptides to stimulate insulin release at concentrations ranging from 0.1 to 10 μM without affecting pancreatic β-cell viability [22,23,24]. 

Preliminary experiments were performed to check the effects of MA1 on cell proliferation and viability. FaO cells were treated for 24 h with 0.1, 1 and 10 µg/mL MA1, in the absence or in the presence of OP. Cell viability was measured by the MTT assay as described in Section 4. The results reported in Figure 1 demonstrated that MA1 did not affect the viability of FaO cells under any of the experimental conditions tested. 

Having been shown to be safe for cell viability, the concentration of 10 μg/mL MA1 and the time interval of 24 h were selected for further experiments. Twenty-four hours was also the optimal incubation interval for investigating the anti-steatotic effects of compounds in this in vitro model [19,20,25].

### 2.2. MA1 Exerts Anti-Steatotic Effects 

The anti-steatotic action of MA1 was assessed by measuring triglyceride (TG) accumulation in FaO cells, and the results are reported in Figure 2. Figure 2A shows that cell exposure to 0.75 mM OP for 3 h resulted in a significant intracellular TG accumulation (+110%; % *p* ≤ 0.001 vs. control), as previously demonstrated [21]. However, the subsequent exposure of steatotic FaO cells to 10 μg/mL MA1 significantly decreased intracellular TG content to the control value (*p* ≤ 0.001 vs. OP). These results were confirmed by fluorescence microscopy visualization of intracellular lipid droplets (LDs) stained in green with BODIPY 493/503 (Figure 2C). As reported in Figure 2B, a few small LDs (about 0.37 μm in size) were dispersed in the cytosol of control cells. In OP-exposed cells, together with small LDs, numerous larger droplets (about 0.92 µm in size; *p* ≤ 0.001 vs. C) were also detectable. After exposure to 10 μg/mL MA1, significant decreases in the number and diameter (about 0.79 μm; *p* ≤ 0.05 vs. OP) of LDs were observed with respect to steatotic cells. TG content in the cell culture medium was also measured as an indication of TG secretion by FaO cells. Figure 2D shows that TG secretion was significantly increased in OP cells with respect to control (+60%, *p* ≤ 0.01), while subsequent MA1 treatment decreased this value to the control level.

### 2.3. MA1 Regulates the Hepatic Expression of PLINs 

The above results indicate an effect of MA1 on LD trafficking in steatotic FaO cells. Therefore, we decided to use qPCR to investigate the expression of genes encoding LD-associated proteins. The perilipin family (PLINs) is the most documented group of such proteins [26], including five members with different functions, of which PLIN2 is the most abundant isoform in the liver. PLIN2 expression is enhanced in NAFLD and correlates with LD abundance [27]. Accordingly, Figure 3 shows that in our model PLIN2 expression was increased in OP cells (2.7-fold increase; *p* ≤ 0.001 vs. C); however, in presence of MA1 we could only observe a decreasing tendency for PLIN2, which did not reach statistical significance. On the other hand, MA1 was capable of enhancing PLIN3 expression in steatotic cells (1.5-fold increase; *p* ≤ 0.001 vs. OP), whereas no significant variations could be appreciated for PLIN5 expression, neither in absence nor in presence of MA1. 

### 2.4. MA1 Regulates the Hepatic Expression of PPARs

As for most of the key genes of hepatic lipid metabolism, PLIN expression is controlled by peroxisome proliferator-activated receptors (PPARs), a family of nuclear transcription factors comprising three members [28], of which PPARα and PPARγ are the most abundantly expressed in FaO cells [29]. Figure 4 shows that in steatotic FaO cells, PPARα expression did not differ significantly from control cells; however, 10 μg/mL MA1 reduced PPARα expression (1.6-fold decrease; *p* ≤ 0.05 vs. C and OP). The mRNA levels of PPARβ/δ were downregulated by OP treatment (1.3-fold decrease; *p* ≤ 0.05 vs. C), but subsequent MA1 administration was ineffective. PPARγ mRNA levels were upregulated in steatotic FaO cells (1.37-fold increase; *p* ≤ 0.05 vs. C), but the subsequent exposure to 10 μg/mL MA1 reduced the expression (2-fold decrease; *p* ≤ 0.05 vs. OP).

## 3. Discussion 

Due to their wide range of biological activities, amphibian peptides represent a new fascinating possibility of finding novel compounds useful for human health. Promising results are achieved in different fields [22,30,31], but their modulating actions on metabolic pathways are of particular interest when considering the global health burden that metabolic diseases represent in Western lifestyle societies. NAFLD is recognized as the leading cause of more severe liver diseases and is linked to other metabolic dysfunctions, such as oxidative stress, endoplasmic reticulum and mitochondria impairments and IR [32] Altogether, these multiple hits [13] lead to the hepatic accumulation of lipids into cytosolic LDs, consisting of a central core of neutral lipids (mainly TGs) surrounded by a monolayer of phospholipids and associated proteins. The LD proteome allows the dynamicity of these organelles, which are considered at the crossroad of lipid trafficking inside the cells [33]. The expression of LD-associated proteins, as well as that of the key genes of hepatic lipid homeostasis, is controlled by specific transcription factors, such as PPARs, and thus different patterns of PPAR expression can be associated with dysregulations of hepatic lipid metabolism in clinical and experimental in vivo and in vitro models [20,34,35]. The anti-oxidant, anti-inflammatory and anti-diabetic properties establish amphibian skin peptides as potential beneficial compounds to be used against fatty liver. We tested this possibility by using a well-established in vitro model of NAFLD, in which we already described PLIN and PPAR expression levels [20,25,36]. In fact, the mRNA levels of both PPARs and PLINs are considered as specific markers in NAFLD models, particularly PPARγ and PLIN2 [20,37].

Our present results demonstrate a strong anti-steatotic effect of MA1, a skin peptide with anti-oxidant and anti-inflammatory properties, which was isolated from the Chinese odorous frog *O. macrotympana* [11]. The anti-steatotic effect of MA1 was evidenced by the decrease in intracellular TG content measured in lipid-overloaded FaO cells. To our knowledge, this is the first time that such a direct anti-steatotic effect has been demonstrated by an amphibian skin peptide. Several amphibian peptides have been evaluated in animal models of high-fat feeding (for a review, see [18]). Some of them showed the ability to significantly reduce body weight, total body fat and circulating triglyceride levels. It is worth noting that our results are in line with this lipid-lowering action, even if no data were reported on the hepatic fat content in these models. Moreover, an insulinotropic activity has been demonstrated for amphibian peptides [18], potentially counteracting fatty liver-associated IR. In addition, anti-oxidant and anti-inflammatory properties have been reported [11]. Should a direct anti-steatotic effect of some amphibian peptides be confirmed by further investigations on experimental in vitro and in vivo models of NAFLD, promising perspectives about the clinical use of these compounds would open. Indeed, the potential ability of amphibian peptides to simultaneously influence several metabolic dysfunctions associated with NAFLD is intriguing, considering that a specific pharmacological treatment against NALFD has not been approved yet. 

In accordance with the diminished TG intracellular content, the number and diameter of LDs, where TGs are stored, were decreased by MA1 treatment, indicating some changes in LD remodeling and trafficking. The increase in PLIN2 expression in OP cells is in accordance with previous findings, demonstrating a correlation between PLIN2 and LD abundance. Consequently, we could expect a decrease in PLIN2 mRNA levels following MA1 treatment. We actually saw a tendency toward downregulation, which deserves further investigation, since it did not reach statistical significance, at least in our hands. The expression of PLIN5 was not significantly affected by OP or MA1 treatments. PLIN5 is known to alleviate palmitic acid-induced inflammatory responses in hepatocytes [38]. However, adding oleate to palmitate-treated cells can fully prevent palmitate lipotoxicity [20,39]. Therefore, a lack of PLIN5 regulation in our experiments is not completely surprising. On the other hand, a novel result is that MA1 was able to increase PLIN3 expression in steatotic FaO cells. PLIN3 (previously known as tail-interacting protein of 47 kDa (TIP47)) is co-expressed with PLIN2 in many tissues, where they often exert overlapping actions on TAG synthesis and LD biogenesis [40]. However, a distinct role of PLIN3 has not been fully clarified yet [41]. A recent paper discovered a new pathway linking PLIN3 and mTOR, in which PLIN3 was essential to activate lipophagy and protect against fatty liver worsening to steatohepatitis [42]. Although our data cannot directly demonstrate that MA1 activates this pathway in steatotic FaO cells, the observed upregulation of PLIN3 by MA1 is in line with this possibility, which should become an interesting line of investigation in future studies. On the contrary, MA1 did not increase extracellular TG content, as measured in cell culture media, indicating that a stimulation of TG secretion by MA1 as a mechanism for intracellular lipid-lowering is unlikely.

Our results also demonstrate a decrease in PPARα and PPARγ expression by MA1. The regulation of PPARs is of great interest because they may serve as potential therapeutic targets for treating NAFLD and metabolic dysfunctions [43]. PPARα is considered the master regulator of hepatic lipid metabolism, particularly during fasting, mainly involved in the regulation of oxidative pathways [44,45]. In vivo, both increased and decreased levels of PPARα expression have been reported in response to high-fat diets [46,47,48]. By using in vitro models, no significant modulations of PPARα expression were observed [20]. These discrepancies can be ascribed to the different experimental conditions utilized to induce hepatic steatosis in vivo and in vitro. This being the case, the downregulation of PPARα expression by MA1 might be associated with a decrease in the activity of lipid oxidative pathways, but the significance of such regulation in steatotic FaO cells is uncertain. On the contrary, PPARγ is a well-recognized marker of liver steatosis that allows energy storage through the activation of lipogenic genes [37,49]. PPARγ expression is increased in lipid-overloaded FaO cells, as previously reported [20]. Accordingly, the anti-steatotic effect of MA1 is coupled with a significant downregulation of PPARγ expression. PPARγ is also a major regulator of PLIN2 expression [50], so the observed tendency of a decrease in PLIN2 expression in response to MA1 could be related to the downregulation of PPARγ mRNA in MA1-treated cells.

We are aware of the need for further studies able to investigate the levels of proteins actually present in steatotic cells and to gain insights into the mechanism of action of MA1. Hopefully, our promising results will pave the way for the application of MA1 as an anti-steatotic compound. 

## 4. Materials and Methods

### 4.1. Materials 

Unless otherwise indicated, all materials and chemicals were supplied by Sigma-Aldrich Corp. (Milan, Italy). All the reagents were of analytical or cell culture grade (≥95% conforming to ACS specifications).

### 4.2. MA1 Synthesis 

MA1 (FLPGLECVW) peptide was manually synthesized using the standard method of solid-phase peptide synthesis according to the 9-fluorenylmethoxycarbonyl (Fmoc) strategy with some modifications [51]. Briefly, Wang-Trp (Boc) deprotected resin was treated with a coupling reaction mixture of 5 equivalents (eq.) of the appropriate Fmoc-amino acid, 4.5 eq. of O-benzotriazol-N,N,N′,N′-tetramethyluroniumhexafluorophosphate (HBTU) and 5 eq. of N,N-diisopropilethylamine (DIPEA) at 0.2 M amino acid final concentration in anhydrous N-methylpyrrolidone (NMP). A solution of 20% (*v/v*) piperidine in DMF was used to remove the Fmoc group. The resin was incubated with a cocktail of acetic anhydride, sym-collidine and DMF (0.5:0.5:9 ratio (*v/v*)) to block the remaining amino groups on the resin. 

The resin was rinsed with dichloromethane and dried. The final cleavage of the peptide from the solid support and removal of all protecting groups were carried out with a trifluoroacetic acid (TFA): tri-isopropylsilane:H_2_O:DODT (90:2.5:2.5:5) mixture. The peptide was precipitated in ice-cold diethyl ether and then purified by preparative reverse phase high-performance liquid chromatography (RP-HPLC) on an Agilent 1260 Infinity preparative HPLC instrument equipped with a Kinetex C18 column (21.20 mm × 150 mm). The separation was obtained with a gradient starting with 5% solvent B for 5 min, linearly increasing to 70% solvent B in 30 min and up to 100% B in 10 min. The solvents used were 0.1% formic acid in water (A) and 0.1% formic acid in acetonitrile (B). The peaks of interest were then collected and evaporated under a vacuum, before being lyophilized and stored at 4 °C. The identity and purity of the MA1 peptide were verified by RP-HPLC, and the collected peak was analyzed in off-line mass spectrometry analysis using an Agilent 1100 series LC/MSD ion trap instrument. The chromatogram and MS spectra are available as Appendix A.

### 4.3. Cell Culture and Treatments 

FaO rat hepatoma cell line (European Collection of Authenticated Cell Cultures, ECACC, Salisbury, Wiltshire, UK) was grown in Coon’s modified Ham’s F-12 medium supplemented with L-glutamine and 10% fetal bovine serum (FBS). 

Cells were incubated in a humidified atmosphere with 5% CO_2_ at 37 °C. For treatments, cells were grown until 80% confluence and then incubated overnight in serum-free medium with 0.25% bovine serum albumin. To induce intracellular lipid accumulation, cells were treated for 3 h with a mixture of oleate/palmitate (OP) at a final concentration of 0.75 mM (2:1 molar ratio) [20]. Thereafter, cells were incubated for 24 h either in control medium (referred to as OP steatotic cells) or in the presence of MA1 at different concentrations. For cell treatment, lyophilized MA1 was dissolved in DMSO at 10 mg/mL and then diluted at 1 mg/mL in serum-free culture medium (stock solution). Serial working dilutions were performed in culture medium. Untreated cells were referred to as controls.

### 4.4. MTT Assay for Determination of Cell Viability 

Cell viability was checked by performing the MTT (3-(4,5-dimethylthiazol-2-yl)-2,5-diphenyltetrazolium bromide) assay. MTT was dissolved in PBS at a concentration of 5 mg/mL and filtered through 0.22 µm pores. The working solution was diluted in culture medium at the final concentration of 0.5 mg/mL. The cells were then incubated for 3 h at 37 °C. Precipitated formazan was then dissolved in acid–alcohol (0.04 N HCl in 2-propanol) solution and read at 570 nm in a Varian Cary-50Bio spectrophotometer (Agilent, Milan, Italy) [21,36].

### 4.5. Quantification of Triglycerides (TGs)

Intracellular TG content was measured in FaO cells using the “Triglycerides liquid” kit (Sentinel, Milan, Italy), as previously described [25,36]. Absorbance was recorded at 546 nm in a Varian Cary-50Bio spectrophotometer (Agilent, Milan, Italy). For the measurement of extracellular TG content, the culture media were processed according to the same method. Values were normalized to protein content as measured by Bradford assay [52], and data are expressed as percent TG content relative to controls. 

### 4.6. Fluorescence Microscopy 

For intracellular lipid staining, cells grown directly on collagen-coated coverslips were rinsed with PBS and fixed with 4% paraformaldehyde for 20 min at room temperature and then quenched with 30 mM NH_4_Cl for 10 min and stained with 250 μg mL^−1^ BODIPY 493/503 (Molecular Probes, Life Technologies, Monza, Italy) for 20 min without permeabilization. After washing, nuclei were stained with 4′,6-diamidino-2-phenylindole (DAPI, 5 μg/mL, (ProLong Gold medium with DAPI; Invitrogen). Images were captured under oil using an Olympus IX70 inverted microscope with a 63x plan apochromatic objective and processed with Adobe Photoshop CS5 [53].

### 4.7. RNA Extraction and Quantitative Real-Time PCR

Total RNA was extracted by using Trizol Reagent (Sigma-Aldrich, Oakville, ON, Canada, and St. Louis, MO, USA) according to the manufacturer’s instructions. One microgram of cDNA was synthesized using RevertAid H-Minus M-MuLV Reverse Transcriptase (Fermentas, Hannover, MD, USA) as previously explained [54]. Real-time quantitative (qPCR) reactions were performed in triplicate in a final volume of 25 μL using 1× SybrGreen SuperMix and Chromo4TM System apparatus (Biorad, Monza, Italy) as previously described [55]. Primer pairs for the genes under analysis (Table 1) were designed ad hoc and synthesized by TibMolBiol custom oligo synthesis service (Genova, Italy). Amplification conditions were as follows: 3 min at 95 °C, followed by 5 s at 95 °C and 1 min at 60 °C for 40 cycles. A melting curve of qPCR products (65–94 °C) was also performed to ensure the absence of artifacts. The relative quantity of target mRNA was calculated by the comparative Cq method using glyceraldehyde 3-phosphate dehydrogenase (GAPDH) as a housekeeping gene and expressed as fold change with respect to controls [56].

### 4.8. Statistical Analysis

Data are means ± S.D. of at least three independent experiments. Statistical analysis was performed using ANOVA with Tukey’s post hoc test (GraphPad Software, Inc., San Diego, CA, USA).

## 5. Conclusions

For the first time, the anti-steatotic action of an amphibian skin peptide, namely macrotympanain A1 (MA1- FLPGLECVW), has been demonstrated in a well-established in vitro model of hepatic steatosis. MA1 is able to decrease intracellular TG content, which results in a reduction in LD number and size. The effect is associated with a downregulation of the expression of PPARγ, which is a marker of hepatic steatosis, and the upregulation of PLIN3, which is known to be involved in the activation of lipophagy. Together with literature evidence documenting anti-oxidant, anti-inflammatory and anti-diabetic effects, our results disclose the possibility of further investigating amphibian skin peptides as a new class of compounds able to reduce hepatic steatosis.

## Figures and Tables

**Figure 1 molecules-27-07417-f001:**
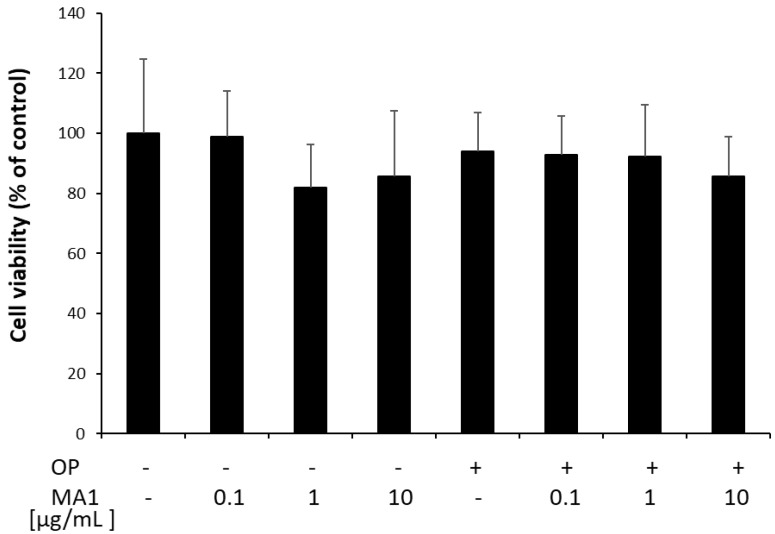
Viability of FaO cells treated for 24 h with 0.1, 1 and 10 µg/mL MA1, in the absence or in the presence of 0.75 mM OP. Data are expressed as percentage of control. Values are mean ± SD from three independent experiments.

**Figure 2 molecules-27-07417-f002:**
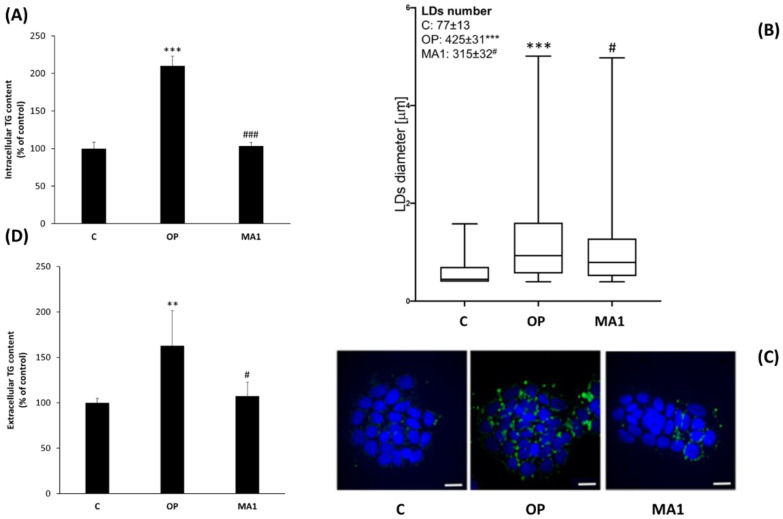
Effects of MA1 on lipid accumulation in steatotic FaO cells. (**A**) Intracellular TG content in control (C) and steatotic cells incubated in the absence (OP) or in the presence of 10 μg/mL MA1 for 24 h. Data are mean ± SD of three independent experiments and are expressed with respect to controls. Significant differences are denoted by symbols on bars (*** *p* ≤ 0.001 vs. C; ### *p* ≤ 0.001 vs. OP). (**B**) Measurements of LD diameter and number in 40 cells for each treatment. The measures of diameters of all counted LDs were plotted as a box-and-whisker plot, showing the interquartile range and the median as a horizontal bar. The whiskers are the minimum and maximum values. The mean number of LDs is reported in the inset table. Significant differences are denoted by symbols (*** *p* ≤ 0.001 vs. C; # *p* ≤ 0.05 vs. OP). (**C**) Representative images of 3 independent experiments showing BODIPY (green)/DAPI (blue) staining of FaO cells to visualize cytosolic LDs (magnification 63×; scale bar: 10 µm). (**D**) Extracellular TG content as measured in the culture medium of the same cells as in (**A**). Data are mean ± SD of three independent experiments and are expressed with respect to controls. Significant differences are denoted by symbols on bars (** *p* ≤ 0.01 vs. C; # *p* ≤ 0.05 vs. OP).

**Figure 3 molecules-27-07417-f003:**
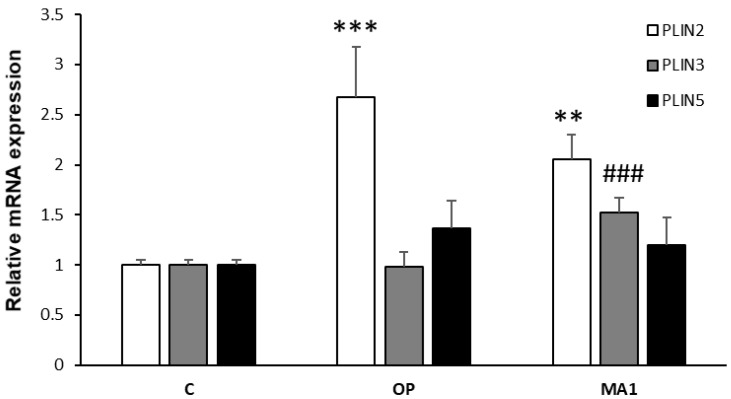
Effects of MA1 on PLIN expression in steatotic FaO cells. Relative expression of PLINs as evaluated by real-time PCR. Steatotic FaO cells (OP) were then incubated for 24 h in the presence of 10 μg/mL MA1. Data are mean ± SD of three independent experiments and are expressed with respect to controls. Significant differences are denoted by symbols on bars (** *p* ≤ 0.01, *** *p* ≤ 0.001 vs. control; ### *p* ≤ 0.001 vs. OP).

**Figure 4 molecules-27-07417-f004:**
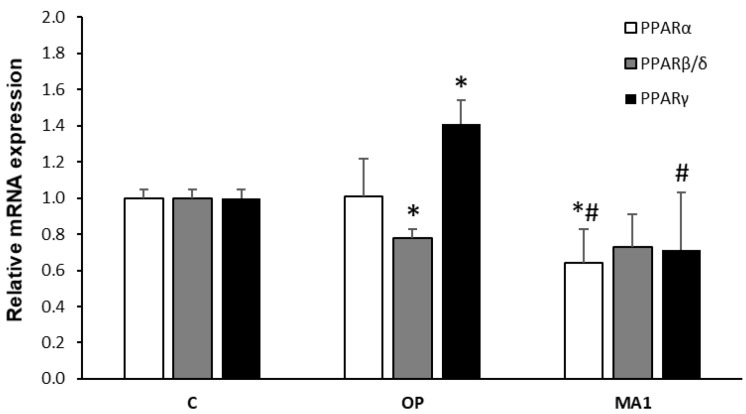
Effects of MA1 on PPAR expression in steatotic FaO cells. Relative expression of PPAR isoforms as evaluated by real-time PCR. Steatotic FaO cells (OP) were then incubated for 24 h in the presence of 10 μg/mL MA1. Data are mean ± SD of three independent experiments and are expressed with respect to controls. Significant differences are denoted by symbols on bars (* *p* ≤ 0.05 vs. control; # *p* ≤ 0.05 vs. OP).

**Table 1 molecules-27-07417-t001:** Primer pairs used for RT-qPCR analysis.

Primer Name	Primer Sequence (5′→3′)	Annealing T (°C)	Product Length (bp)	Accession ID
GAPDH Fwd	GACCCCTTCATTGACCTCAAC	60	136	DQ403053
GAPDH Rev	CGCTCCTGGAAGATGGTGATGGG			
PPARα Fwd	CCCCACTTGAAGCAGATGACC	60	139	NM_013196
PPARα Rev	CCCTAAGTACTGGTAGTCCGC			
PPARβ/δ Fwd	AATGCCTACCTGAAAAACTTCAAC	60	96	AJ306400.1
PPARβ/δ Rev	TGCCTGCCACAGCGTCTCAAT			
PPARγ Fwd	CGGAGTCCTCCCAGCTGTTCGCC	60	116	Y12882
PPARγ Rev	GGCTCATATCTGTCTCCGTCTTC			
PLIN2 Fwd	CCGAGCGTGGTGACGAGGG	60	148	AAH85861
PLIN2 Rev	GAGGTCACGGTCCTCACTCCC			
PLIN3 Fwd	GGAACTGGTGTCATCAACAG	60	108	NW_047865.1
PLIN3 Rev	GGTCACATCCACTGCTCCTG			
PLIN5 Fwd	GGATGTCCGGTGATCAGAC	60	96	XM_576698
PLIN5 Rev	GTGCACGTGGCCCTGACCAG			

## Data Availability

The data presented in this study are available in this article.

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
