# Peer review of "First Evidence of Anti-Steatotic Action of Macrotympanain A1, an Amphibian Skin Peptide from Odorrana macrotympana"

_molecules, 2022, doi:10.3390/molecules27217417_

Round 1

Reviewer 1 Report

Figure 1. The purpose of measuring the cell viability in this experimental design. How were the three concentrations given chosen? What were the experimental design and conditions? Did the peptide alone was tested? Why have not considered IC50? If considered, what was the value of IC50? Any other reference sources with regard to this experiment?

Figure 2. Why it was chosen to treat 10 μg/mL MA1? How did you determine the optimum time of 24h? Provide statistical analysis method. Can the p values for each experiment be provided? Especially in 2D, does the OP and MA1 statistically significant? 2C is a representation of how many independent experiments (3)? How many LD diameters were taken into account in each independent experiment?

Why did the authors choose to investigate PLIN levels by mRNA levels?

How would you address the different expressions of different PLIN levels in OP and MA1?

How would you correlate PLIN levels to PPARγ levels?

Any data can be provided for the LC/MSD for peptide synthesis? Any citations related to this?

The discussion lacks significant discussion between your results and previous findings and how your research could stand out or bring forward novel findings, as explained in the title.

Author Response

We wish to thank all referees for their time and effort in reviewing our work.

The manuscript has undergone extensive revision taking into account all the issues raised by the reviewers. Fig. 1 and 2 have been modified and the Discussion section has been enriched with the points raised by the reviewers, together with appropriate references.

We believe that our work has improved and hope it is acceptable for publication on Molecules.

REVIEWER 1

Figure 1.

Q: How were the three concentrations given chosen?

A: The concentration used were chosen on the basis of previous studies that investigated the insulinotropic properties of several amphibian peptides using in vitro models. This point has been added to the first paragraph of the Results section, together with appropriate references.

Q: What were the experimental design and conditions?

A: The experimental design and conditions have been detailed in the first paragraph of the Results section, and the technical description is under Materials and Methods: in the revised version, we dedicated a separate paragraph to “MTT Assay for Determination of Cell Viability” (paragraph 4.4).

Q: Did the peptide alone was tested?

A: The peptide alone was tested and did not affect cell viability. The results have been added to Figure 1.

Q: Why have not considered IC50? If considered, what was the value of IC50?

A: Thank you for the comment, we apologize for not being clear. We did not consider IC50 since we did not observe direct effects of MA1 on cell viability and proliferation. In other words, we did not report any cytotoxic activity for the concentrations tested. It was not our goal to investigate the cytotoxicity of MA1. Instead, we were interested in beneficial effects, so we only checked a concentration range suitable for this purpose on the basis of literature evidence. The 2.1 paragraph of the Results section has been completely re-arranged for the sake of clarity.

Q: Any other reference sources with regard to this experiment?

A: The reference to the method is in “MTT Assay for Determination of Cell Viability”, paragraph 4.4 of the Materials and Methods section. We routinely measure cell viability of FaO cells when using new compounds on the model, so we also added the reference to a recent paper of ours.

Figure 2.

Q: Why it was chosen to treat 10 μg/mL MA1?

A: Since the maximal concentration tested was not detrimental to steatotic FaO cells, we carried on with this dose, which in any case was in line with the concentrations used in vitro models to demonstrate the insulinotropic properties of amphibian peptides. These considerations have been added to the text, in the first paragraph of the Results section.

Q: How did you determine the optimum time of 24h?

A: The time was determined because we didn’t see a decline in cell viability and on the basis of our previous studies on anti-steatotic compounds in the FaO model. We explained and added references for this in the first paragraph of the Results section.

Q: Provide statistical analysis method. Can the p values for each experiment be provided?

A: The statistical analyses are indicated in the 4.7 paragraph “Statistical Analysis”, under the Materials and Methods section. The p value is always indicated, please refer to figure captions when not indicated in the text.

Q: Especially in 2D, does the OP and MA1 statistically significant?

 A: Yes, the values of OP and MA1 are significantly different, as denoted by the # symbol on bar and reported in the figure caption (# p≤0.05 vs OP). On the contrary, MA1 was not significantly different from C.

Q: 2C is a representation of how many independent experiments (3)?

A: Yes, 3 independent experiments, as we added in the figure caption.

Q: How many LD diameters were taken into account in each independent experiment?

A: Measurements of LDs diameter and number were done in 40 cells for each treatment, as indicated in figure 2 caption. The inset table in Fig. 2B reports the mean of counted LDs in 3 independent experiments: we added SD values. The diameters were measured in all LDs in each experimental condition. We hope that the figure caption is now clearer.

Q: Why did the authors choose to investigate PLIN levels by mRNA levels?

A: We have extensive experience in evaluating the mRNA expression profile of PLINs and PPARs: please, see our review  (doi: 10.5604/01.3001.0010.2713) and recent examples of our previous work (doi: 10.1016/j.lfs.2022.120468; doi: 10.3390/molecules26154467; doi: 10.3390/molecules26041161;  doi: 10.3389/fphys.2015.00418). Undoubtedly, the measurements of protein levels would represent additional information to our results but, unfortunately, we are unable to carry out such experiments in a reasonable time. Nevertheless, the mRNA expression levels of both PPARs and PLINs are considered as specific markers in NAFLD models, particularly PPARγ and PLIN2. We have acknowledged these issues in the Discussion section.

Q: How would you address the different expressions of different PLIN levels in OP and MA1?

A: Thank you for the comments, the Discussion section has been expanded on this point.

Q: How would you correlate PLIN levels to PPARγ levels?

A: Our results revealed a regulation of PLIN2 and PLIN3. PLIN2 is a known target gene of PPARγ, while less is known about PLIN3. We added some comments in the Discussion.

Q: Any data can be provided for the LC/MSD for peptide synthesis? Any citations related to this?

A: We corrected the Methods: “The identity and purity of the MA1 peptide was verified by RP-HPLC and the collected peak was analyzed in off-line mass spectrometry analysis using an Agilent 1100 series LC/MSD ion trap instrument”. The chromatogram and MS spectra are provided as Supplementary materials (Figure S1).

Q: The discussion lacks significant discussion between your results and previous findings and how your research could stand out or bring forward novel findings, as explained in the title.

A: To our knowledge, this is the first time that an amphibian peptide was tested for a direct anti-steatotic effect. So, we are actually presenting a new potential application for the antimicrobial peptide MA1. Anti-diabetic and anti-obesity effects have been reported for several amphibian peptides. The Discussion section has been expanded on this point, but the most novel finding remains the direct anti-steatotic effect.

REVIEWER 2

In the manuscript entitled "First evidence of anti-steatotic action of Macrotympanain A1, an amphibian skin peptide from Odorrana macrotympana" the authors are presenting a new application for the antimicrobial peptide MA1.

The application as anti-steatotic compound is for the first time described for antimicrobial peptides, with encouraging results.

We thank the reviewer for appreciating our work and the novelty of our study.

Overall the manuscript is well done and the data show significant effects, however, there are some parts that need improvement. 

Q: Although the OP term is explained in material and methods section, the first time is seen in the manuscript is in section 2, the results. The authors should explain it also in section 2.1. to make it easier for the reader. 

A: Thank you for the comment. We have re-arranged the first paragraph of the Results section to better explain the experimental design and make it clearer to the reader. The technical description of the first experiment is under Materials and Methods: in the revised version, we dedicated a separate paragraph to “MTT Assay for Determination of Cell Viability” (paragraph 4.4).  

Q: Also the graph 1 is not sufficiently described. The authors should explain the experimental conditions. Did the peptide alone was tasted? 

A: The experimental conditions are now clarified in the re-arranged first paragraph of the Results section. The peptide alone was tested but did not affect cell viability, so we  We added the bars in the new fig 1.

Q: The images in figure 2C are too small. They should be larger in order to see better the cytosolic LDs.

A: The images in figure 2C have been enlarged.

Reviewer 2 Report

In the manuscript entitled "First evidence of anti-steatotic action of Macrotympanain A1, an amphibian skin peptide from Odorrana macrotympana" the authors are presenting a new application for the antimicrobial peptide MA1.

The application as anti-steatotic compound is for the first time described for antimicrobial peptides, with encouraging results.

Overall the manuscript is well done and the data show significant effects, however, there are some parts that need improvement. 

Although the OP term is explained in material and methods section, the first time is seen in the manuscript is in section 2, the results. The authors should explain it also in section 2.1. to make it easier for the reader. 

Also the graph 1 is not sufficiently described. The authors should explain the experimental conditions. Did the peptide alone was tasted? 

The images in figure 2C are too small. They should be larger in order to see better the cytosolic LDs.

Author Response

We wish to thank all referees for their time and effort in reviewing our work.

The manuscript has undergone extensive revision taking into account all the issues raised by the reviewers. Fig. 1 and 2 have been modified and the Discussion section has been enriched with the points raised by the reviewers, together with appropriate references.

We believe that our work has improved and hope it is acceptable for publication on Molecules.

REVIEWER 2

In the manuscript entitled "First evidence of anti-steatotic action of Macrotympanain A1, an amphibian skin peptide from Odorrana macrotympana" the authors are presenting a new application for the antimicrobial peptide MA1.

The application as anti-steatotic compound is for the first time described for antimicrobial peptides, with encouraging results.

We thank the reviewer for appreciating our work and the novelty of our study.

Overall the manuscript is well done and the data show significant effects, however, there are some parts that need improvement. 

Q: Although the OP term is explained in material and methods section, the first time is seen in the manuscript is in section 2, the results. The authors should explain it also in section 2.1. to make it easier for the reader. 

A: Thank you for the comment. We have re-arranged the first paragraph of the Results section to better explain the experimental design and make it clearer to the reader. The technical description of the first experiment is under Materials and Methods: in the revised version, we dedicated a separate paragraph to “MTT Assay for Determination of Cell Viability” (paragraph 4.4).  

Q: Also the graph 1 is not sufficiently described. The authors should explain the experimental conditions. Did the peptide alone was tasted? 

A: The experimental conditions are now clarified in the re-arranged first paragraph of the Results section. The peptide alone was tested but did not affect cell viability, so we  We added the bars in the new fig 1.

Q: The images in figure 2C are too small. They should be larger in order to see better the cytosolic LDs.

A: The images in figure 2C have been enlarged.

Round 2

Reviewer 2 Report

The authors have made the changes asked and the paper can be published in the present form